# AFBench: A Large-scale Benchmark for Airfoil Design

**Jian Liu**[1,2]  **Jianyu Wu**[2]  **Hairun Xie**[3]  **Guoqing Zhang**[1]  **Jing Wang**[3]  **Wei Liu**[3]
**Wanli Ouyang**[2]  **Junjun Jiang**[1]  **Xianming Liu**[1*]  **Shixiang Tang**[2*]  **Miao Zhang**[3*]

[1] Harbin Institute of Technology    [2] Shanghai Artificial Intelligence Laboratory
[3] Shanghai Aircraft Design and Research Institute

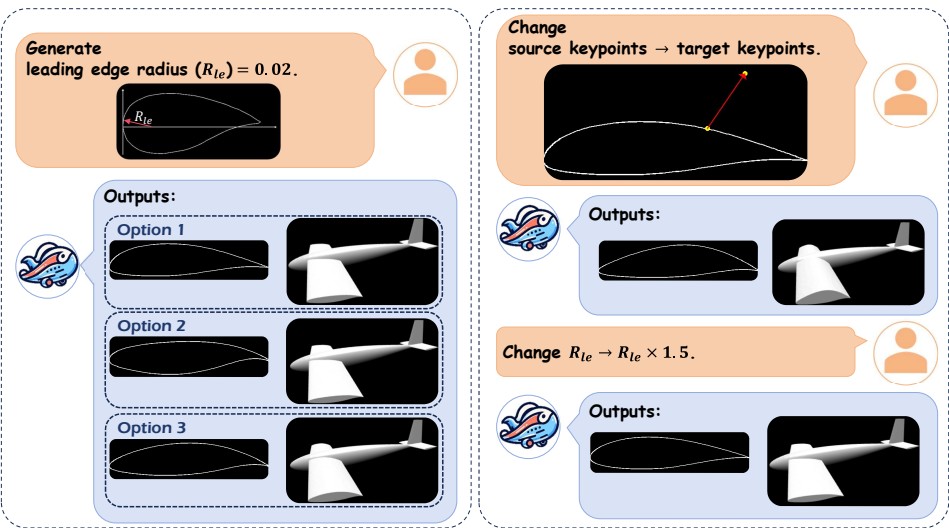

Figure 1: **Our Airfoil Generation and Editing Software.** (a) Generating diverse candidate airfoils. (b) Editing Keypoints and Editing Physical Parameters.

## Abstract

Data-driven generative models have emerged as promising approaches towards achieving efficient mechanical inverse design. However, due to prohibitively high cost in time and money, there is still lack of open-source and large-scale benchmarks in this field. It is mainly the case for airfoil inverse design, which requires to generate and edit diverse geometric-qualified and aerodynamic-qualified airfoils following the multimodal instructions, *i.e.,* dragging points and physical parameters. This paper presents the open-source endeavors in airfoil inverse design, *AFBench*, including a large-scale dataset with 200 thousand airfoils and high-quality aerodynamic and geometric labels, two novel and practical airfoil inverse design tasks, *i.e.,* conditional generation on multimodal physical parameters, controllable editing, and comprehensive metrics to evaluate various existing airfoil inverse design methods. Our aim is to establish *AFBench* as an ecosystem for training and evaluating airfoil inverse design methods, with a specific focus on data-driven controllable inverse design models by multimodal instructions capable of bridging the gap between ideas and execution, the academic research and industrial applications. We have provided baseline models, comprehensive experimental observations, and analysis to accelerate future research. Our baseline model is trained on an

---

*Corresponding Author

RTX 3090 GPU within 16 hours. The codebase, datasets and benchmarks will be available at https://hitcslj.github.io/afbench/.

# 1   Introduction

The airfoil inverse design problem serves as the center of the automatic airfoil design, which is to seek design input variables, *i.e.,* physical parameters, to optimize an underlying objective function, *e.g.,* aerodynamics. Previous methods can be divided into two categories: optimization methods [1, 2, 3] and data-driven methods [4, 5, 6, 7]. First, the optimization-based methods usually design an objective function by constructing a mathematical model and leverage the typical optimization algorithms, *e.g.,* genetic algorithms [8], adjoint optimization [9] and topology optimization [10], to find the optimal input variables as the design parameters. Despite the success, these methods have limitations in considerable time consumption and the diversity of the optimal design variables due to the constructed physical model of airfoils. Second, the data-driven methods [11, 12, 13, 14] typically borrow ideas from the advancements in conditional generative models in artificial intelligence. Popular generative methods such as CGAN [15], CVAE [16], and Diffusion models [17, 18] have been explored, demonstrating their effectiveness. However, current data-driven methods suffer from the following three drawbacks. First, the existing datasets are relatively small-scale, *e.g.,* the design geometry dataset UIUC [19] contain only thousands of samples. Therefore, data-driven models trained on such datasets have limited generalization capabilities and fail to generate diverse solutions that meet the requirements. Second, the current datasets typically provide only a single condition, *i.e.,*aerodynamic parameters, and thus cannot handle multi-condition design, *i.e.,* controlling leading edge radius and upper crest position as geometric parameters simultaneously, which are real industrial applications in airfoil design. Third, current airfoil inverse design methods do not support progressive editing existing designs according to manual and multimodal requirements, which limits their applications in the industry. For example, one of our authors from Shanghai Aircraft Design and Research Institute, who has over 10 years experiences for airfoil design, claimed that each airfoils used in current commercial airplanes underwent years of progressively refinements by hundreds of engineers.

To drive the development of generative models in the field of engineering design, we construct a comprehensive airfoil benchmark, *AFBench*, that can be a cornerstone to cope with the aforementioned challenges with the following merits:

**(1) Tasks – Multi-Conditional Generation and Editing in Airfoil Inverse Design:** Regarding the aforementioned dataset, we tailor it to accommodate two new but more practical tasks in real airfoil design: multi-conditional airfoil generation and multimodal airfoil editing. The task of airfoil generation is not limited to the previous approach of generating airfoils based solely on given aerodynamic labels such as Lift-to-drag ratio. Instead, it involves generating airfoils based on multiple intricate geometric labels proposed by our authors who are experts in airfoil designs, which is more challenge but practical than previous airfoil generation based on the single condition. We introduce a newly developed airfoil editing task, which currently allows for the modification of both control points and physical parameters of the airfoil. The editing of physical parameters is not present in traditional airfoil editing software, and the movement of control points, compared to the spline interpolation in traditional software, is enabled by AI models with a broader design space.

**(2) Datasets - Large-scale Airfoil Datasets with High-quality and Comprehensive Geometric and Aerodynamic Labels:** Regarding the aforementioned airfoil inverse design tasks, the training subset of the proposed AFBench consists of 200,000 well-designed both synthetic and manually-designed airfoils with 11 geometric parameters and aerodynamic properties under 66 work conditions (Mach number from 0.2 to 0.7, Lift coefficient from 0 to 2). To construct the AFBench, we propose an automatic data engine that includes data synthesis, high-quality annotations and low-quality filtering. Different from previous datasets, we (i) not only combine all airfoils in the existing datasets such as such as UIUC [19] and NACA [20], but also include 2,150 new manually-designed supercritical airfoils from Shanghai Aircraft Design and Research Institute that is highly insufficient in existing datasets; (ii) further enlarge the dataset to 200,000 airfoils by effective data synthesis with conventional physical models and unconditional generative models; (iii) annotate geometric and aerodynamic labels by CFD (Computational fluid dynamics) simulation software.

**(3) Open-source Codebase and Benchmarks – A Open-source Codebase of Data-driven Generative Models for Airfoil Inverse Design with State-of-the-art Methods and Comprehensive**

**Evaluation Metrics:** Since there are still absent a comprehensive and clean codebase to compare and analyze different airfoil inverse design methods, we release a comprehensive and publicly accessible codebase to facilitate future researches. This codebase includes multiple existing methods, *e.g.,* cVAE [21, 22, 23], cGAN [24, 25], and our newly proposed primary architectures for both multi-conditional airfoil generation and controllable airfoil editing, PK-VAE, PK-VAE$^2$, PK-GAN, PKVAE-GAN, PK-DIFF, PK-DiT inspired by mainstream generative frameworks, *i.e.,* VAE, GAN and Diffusion models. To facilitate exploration and usage, we have also provided a user-friendly demo that easily allows different airfoil inverse design methods for online generation and editing. Furthermore, different from previous benchmarks that only evaluates the aerodynamic performance, we also provide the interface to evaluate the geometric quality, the aerodynamic quality and the diversity of the generated airfoils, which are also crucial for airfoil inverse design.

The main contributions of this work are summarized as follows:

- We propose the use of generative methods for two key tasks in airfoil design: multi-conditional airfoil generation and airfoil editing. We also establish comprehensive evaluation metrics including diversity, controllability, geometric quality and aerodynamic quality.

- We propose a large-scale and diverse airfoil dataset in Airfoil Generative Design. This dataset includes 200 thoushands airfoil shapes, accompanied by detailed geometric and aerodynamic annotation labels. The dataset can provide valuable resources for training and evaluating generative models in airfoil inverse design.

- We construct and open-source a codebase that encompasses generative methods in airfoil design, including foundational techniques such as cVAE, cGAN as well as advanced models like PK-GAN,PK-VAE,PKVAE-GAN and PK-DiT. We provide a user-friendly demo that allows for visualizing and experiencing airfoil design in real-time.

## 2   Related Work

**Airfoil Inverse Design.** The ultimate goal of airfoil inverse design is to use algorithms to automatically find airfoils that meet the given requirements. Previous efforts [26, 27] have explored datasets for investigating airfoil aerodynamic characteristics, but have largely relied on the UIUC and NACA airfoil shapes, lacking the support to explore large-scale airfoil models. Our dataset, on the other hand, boasts a more diverse collection of airfoils and rich annotations. Additionally, we have proposed AFBench, which includes airfoil generation and airfoil editing. Airfoil generation is a combination and complement to inverse design and parameterization, as it can generate airfoils that satisfy geometric constraints based on PARSEC parameters. We also propose a new task, airfoil editing, to allow designers to more easily find the optimal airfoil based on their experience.

**Conditional Generative methods in Airfoil Design.** There are some new attempts to leveraging the advantages of both implicit representation and generative methods in airfoil design. *Variation Auto Encoder* [28] trains a model to minimize reconstruction loss and latent loss, and it is usually optimized considering the sum of these losses. [22] proposes two advanced CVAE for the inverse airfoil design problems by combining (CVAE) and distributions. *Generative adversarial network* [29] uses a generative neural network to generate a airfoil and uses a discriminative neural network to justify the airfoil is real or fake. CGAN [15] improves the original GAN by inputting the conditions to both the generator and discriminator. For instance, [30] generates shapes with low or high lift coefficient. By inputing the aerodynamic characteristics such as lift-to-drag ratio (Cl/Cd) or shape parameters, it is possible to guide the shape generation process toward particular airfoil. *Diffusion model* [17] is the emerging generative models [14, 31, 13] in engineering design. However, there are still few attempts in airfoil inverse design. We provide more detailed literature review in Appendix B.

## 3   Automatic Data Engine

Since diverse airfoil datasets are not easily accessible publicly, we develop a data engine to collect 200,000 diverse airfoils, dubbed AF-200K. Our proposed AF-200K dataset first includes airfoils from two public datasets, such as UIUC and NACA, and then leverages our proposed data engine to generate synthetic airfoils. The data engine has three stages: (1) synthetic airfoil generation stage; (2) geometric and aerodynamic parameter annotation stage; (3) low-quality airfoil filtering stage. We illustrate the data engine pipeline in Fig. 2 and visualize generated airfoils in Fig. 3.

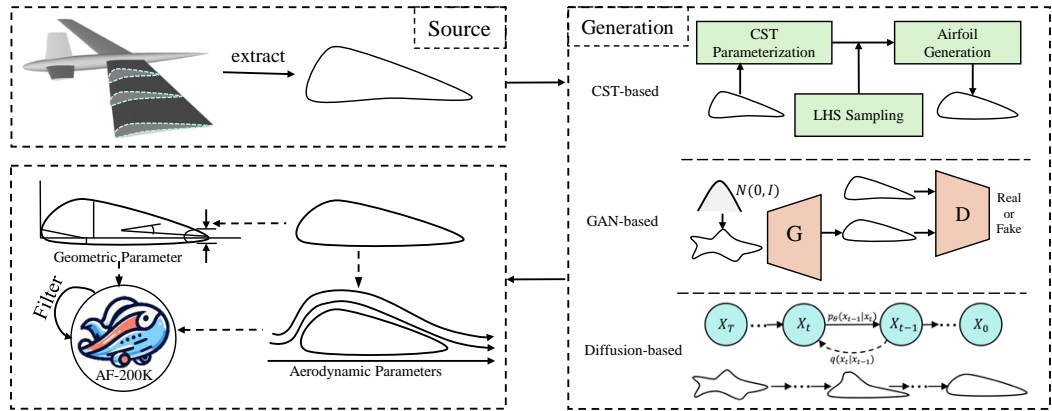

Figure 2: **The overall pipeline of the Automatic data engine**

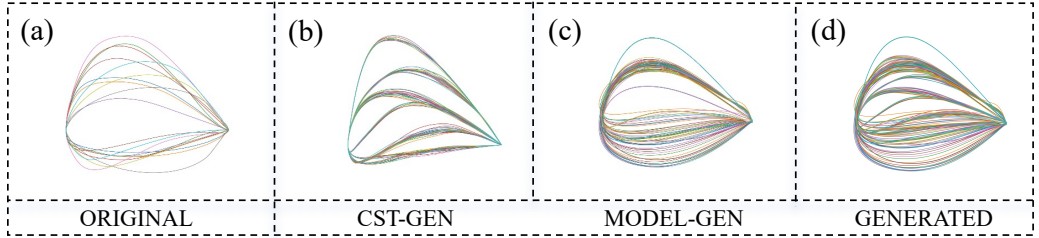

Figure 3: **Diverse Airfoils Generated by Our Automatic Data Engine.** (a) Original airfoil from the UIUC database. (b) Airfoils generated using CST perturbation of the UIUC airfoil. (c) Airfoils generated using our trained model. (d) Combination of generated airfoils.

## 3.1 Synthetic Airfoil Generation Stage

Based on airfoils in UIUC and newly collected airfoils that are manually designed by COMAC (Commercial Aircraft Corporation of China), we synthesize airfoils by both physical models and unconditional generative methods.

**CST-assisted Generation.** The CST-assisted Generation synthesizes the airfoils firstly by parameterizing physical models, *i.e.,* CST model and then perturbing these parameters. Given one manually designed airfoil $f_0$, we parameterize the airfoil with the CST method [32] as $p_0 = (p_0^1, p_0^2, ..., p_0^M)$, where $M$ is the number of physical parameters [2]. Afterwards, we perturb the parameters of the airfoils with Latin hypercube sampling (LHS). Take generating $N$ airfoils based on one manually-designed airfoil for example. For every variable $(p_0^1, p_0^2, ..., p_0^M)$ in our parameterized airfoil, we evenly divided into $N$ parts, and randomly sample a value in $N$ parts, respectively for $N$ generated airfoils. With Latin hypercube sampling (LHS), the generated airfoils can be uniformly sampled from the parametric space of CST for supercritical airfoils. The generated airfoils are illustrated in Fig. 3.

**Unconditional Airfoil Generation Stage.** While the airfoils generated by perturbing the parameters of CST models significantly extend the training datasets, the design space is still limited to the capability of CST models (Fig. 3). To further explore the more general design space, we propose two unconditional generative-model-based methods, *i.e.,* BézierGAN [33] and diffusion models [17], to generate airfoils in the training set. Specifically, we train BézierGAN [33] and diffusion models [17] using our selected airfoils from the UIUC dataset (referred to as UIUC-Picked). We generate 10,000 airfoils with BézierGAN and generate another 10,000 airfoils with the diffusion model. We will detail the architecture of BézierGAN and diffusion model in the Appendix D.

---

[2]The exact formulation of CST models and fitting methods are detailed in the Appendix C.

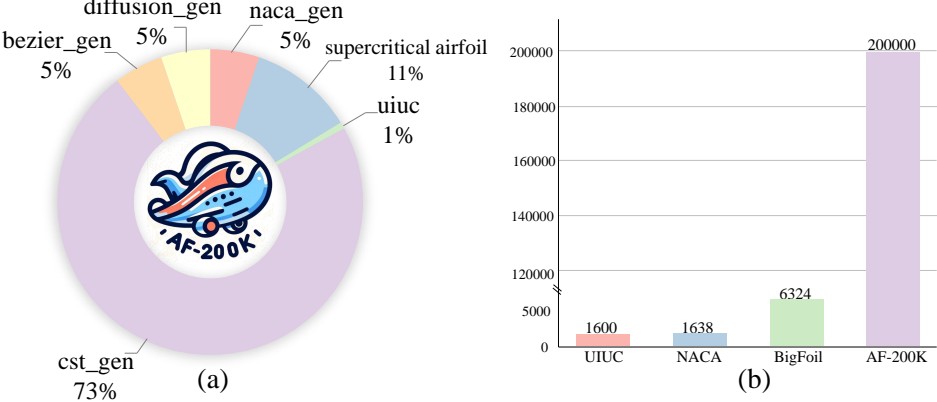

Figure 5: Dataset presentation: (a) Proportional composition of the datasets within AF-200K. (b) Comparison between AF-200K and existing airfoil datasets.

## 3.2 Geometric and Aerodynamic Parameter Annotation Stage

**Aerodynamic Annotation.** We compute the angle of attack (AoA) and drag coefficient (CD) of each airfoil under different working conditions. Specifically, we set the Reynolds number to 100,000 and vary the Mach number from 0.2 to 0.7 (in increments of 0.1), and the lift coefficient (CL) from 0.0 to 2.0 (in increments of 0.2). The working conditions are denoted as $w_c = [Ma, CL]$, where Ma is the Mach number and CL is the lift coefficient. For each working condition, we pass the airfoil coordinates into XFoil [34] to calculate the corresponding aerodynamic labels, including the angle of attack (AoA), drag coefficient (CD), and moment coefficient (CM).

**Geometric Annotation.** The Geometric label is primarily based on PARSEC physical parameters, with Control keypoints as supplementary information. The PARSEC physical parameters (as shown in Fig. 4) include the leading edge radius ($R_{le}$), upper crest position ($X_{up}, Y_{up}$), upper crest curvature ($Z_{xxup}$), lower crest position ($X_{lo}, Y_{lo}$), lower crest curvature ($Z_{xxlo}$), trailing edge position ($Y_{te}$), trailing thickness ($\Delta Y_{te}$), and two trailing edge angles ($\alpha_{te}, \beta_{te}$).

We utilize B-spline [35] interpolation to convert the discrete points into a continuous representation, and then calculate the first-order and second-order derivatives, as well as the extrema. For the Control Keypoints, we select a subset of the airfoil surface points, approximately one-twentieth of the original number of points. The main purpose is to control the overall contour of the airfoil, ensuring that it does not undergo drastic changes.

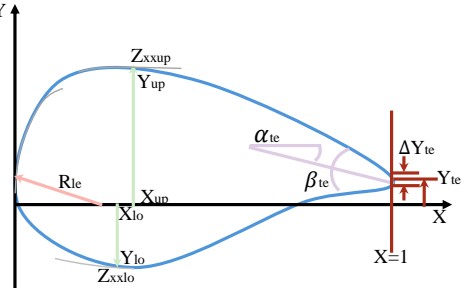

Figure 4: PARSEC physical parameters

## 3.3 Airfoil Filtering Stage

Given the airfoils generated with the parametric CST model and the generative model, we need to filter out those with low aerodynamic performance to prevent the generative model from producing low-quality airfoils. Specifically, we use a numerical solver based on Reynolds-Averaged Navier-Stokes (RANS) equations to calculate the physical parameters of flow fields. These parameters are then used to assess the aerodynamic performance of the generated airfoils. We set 66 work conditions (as detailed in Section 3.2), and if an airfoil fails to converge under all 66 conditions, we classify it as a poor-quality airfoil and discard it.

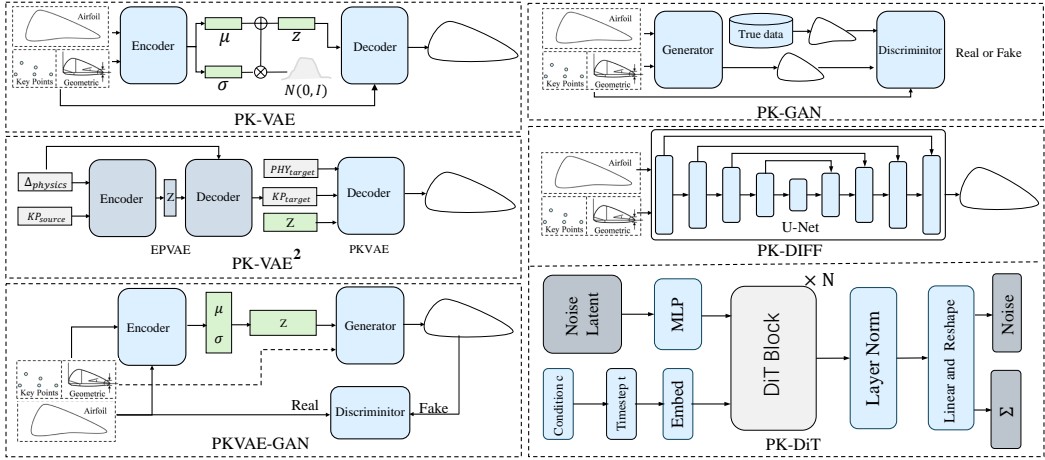

Figure 6: **The baseline methods for benchmarking the dataset.**

## 4  AFBench: Dataset Presentation and Benchmarking Setup

### 4.1  Dataset Presentation

Based on the aforementioned automatic data engine, the AF-200K dataset includes a diverse collection of about 200,000 airfoils, including UIUC, NACA, supercritical airfoils, and generated airfoils, as shown in Fig. 5. From the UIUC dataset, we have carefully selected 1,433 airfoils with favorable aerodynamic performance from more than 1,600 original raw data entries. For the NACA airfoils, we referenced the AIRFRANS [27] design space, resulting in a total of 5,000 NACA 4-digit and 5,000 NACA 5-digit airfoils. The Supercritical Airfoil dataset was generated by perturbing and expanding upon designs provided by COMAC engineers, using the CST method, yielding a total of 21,500 airfoils. To further augment the UIUC dataset, we generated 143,300 airfoils through CST-assisted generation. Additionally, we employed generative modeling approaches to synthesize 10,000 airfoils each using BézierGAN [33] and diffusion models [17]. All airfoil data are stored in the form of 2D coordinates, with each airfoil represented by 257 points. In cases where the original data did not have 257 points, we used B-spline interpolation to ensure a consistent representation. The AF-200K dataset is split into training, validation, and test sets with a ratio of 8:1:1.

### 4.2  Airfoil Inverse Design Tasks

**Controllable Airfoil Generation.** The controllable airfoil task aims at generating airfoils, which is described by 257 points, given the physical parameters and control keypoints. We expect the generated airfoil should consist with the given physical parameters and control keypoints and also with high diversity, good geometric quality and aerodynamic quality.

**Editable Airfoil Generation.** The editable airfoil generation task aims at editing a given airfoil following the instruction. Specifically, the editable airfoil we can edit the airfoil using the physical parameters and control keypoints, *e.g.,* 2 times enlarging the leading edge, or dragging one of the control point. We expect the airfoil after editting can be conformed to the instruction. We expect the airfoil after editing should be consist with the given physical parameters or control keypoints and also with high diversity, good geometric quality and aerodynamic quality.

### 4.3  Baseline Methods

As shown in Fig. 6, we train four generative models: VAE, GAN, VAE-GAN and Diffusion Models.

**PK-VAE and PK-VAE$^2$.** Based on [36], we modify the plain VAE by incorporating parsec parameters [37] and control keypoints as geometry constraints. PK-VAE$^2$ is a composite of VAEs: EK-VAE, PK-VAE and PK-VAE, which enable airfoil editing. Speicifically, EK-VAE achieves editing by predicting physical parameters from control keypoints, while EP-VAE predicts control keypoints

from physical parameters. By training these components separately and then combining them with PK-VAE for joint training, we can achieve efficient airfoil editing.

**PK-GAN.** Building upon the Bézier-GAN approach [33], we introduce a conditional formulation in our model. We employ a technique similar to Adaptive Instance Normalization [38] to seamlessly integrate the condition embedding at multiple scales within the Generator. Simultaneously, we adopt a similarity-based approach to blend the condition information into the Discriminator.

**PKVAE-GAN.** Inspired by [39], we utilize a conditional Variational Autoencoder (cVAE) as the generator, and train it with the discriminator conditioned on physical parameters and keypoints.

**PK-Diffusion.** For conditional diffusion models [18, 40], we designed two types based on Unet [41] and Transformer [40] architectures. In the U-Net model (PK-DIFF), the encoder extracts features by stacking multiple layers of convolutions, while the decoder reconstructs features by stacking multiple layers of convolutions. The encoder outputs are concatenated with the corresponding scale inputs of the decoder through skip connections. The time steps and conditions are mapped through MLP layers and then integrated with the input features. In the DIT model (PK-DIT), we also integrate time steps and conditions by employing MLP to map them before feature extraction. Feature extraction is performed through four layers of DIT blocks.

## 4.4 Evaluation Metrics

The performance of the model depends on three factors: controllability, diversity of the generated/edited airfoils, and quality of the generated airfoils (including both geometric and aerodynamic quality). We evaluate the performance using the following metrics:

- To measure the constraint of the conditions, we propose the *label error*:

$$\sigma_i = |\hat{p}_i - p_i|, i = 1, 2, ..., 11 \tag{1}$$

where $\sigma_i$ is the label error for the $i$-th physical parameter, $\hat{p}_i$ is the $i$-th physical parameter calculated from the generated airfoil, $p_i$ is the $i$-th physical parameter of the given condition. We denote $\{p_i\}_{i=1}^{11}$ as $\{R_{le}, X_{up}, Y_{up}, Z_{xxup}, X_{lo}, Y_{lo}, Z_{xxlo}, Y_{te}, \Delta Y_{te}, \alpha_{te}, \beta_{te}\}$, respectively.

- To quantify the *diversity* of the generated airfoils, we propose the following formula:

$$\mathcal{D} = \frac{1}{n} \sum_{i=1}^{n} \log det(\mathcal{L}_{S_i}), \tag{2}$$

where $n$ is the number of sample times, and the set of generated airfoils is denoted as $\mathbf{F} = (f_1, f_2, ..., f_M)$. The $i$-th subset of the data, $S_i$, is a subset of $\mathbf{F}$ with a smaller size $N$ (where $N < M$). The matrix $\mathcal{L}_{S_i}$ is the similarity matrix, calculated based on the Euclidean distances between the airfoils in the subset $S_i$, as proposed in [42]. The $det(\mathcal{L}_{S_i})$ represents the determinant of the similarity matrix $\mathcal{L}_{S_i}$, and the $\log det(\mathcal{L}_{S_i})$ is the natural logarithm of the determinant, which is used to prevent numerical underflow.

- To measure the geometric quality of the airfoils, we propose the *smoothness* metric:

$$\mathcal{M} = \sum_{i=1}^{N} \text{Distance}_{Pn\perp|P_{n-1}P_{n+1}|}, \tag{3}$$

where $P_n$ is the $n$-th point, $|P_{n-1}P_{n+1}|$ is the line connecting the adjacent points, and Distance calculates the perpendicular distance from point $P_n$ to the line $|P_{n-1}P_{n+1}|$. $N$ represents the number of generated airfoils.

- To measure the aerodynamic quality of the model, we propose the *success rate*. We generate airfoils and evaluate whether they converge under M different work conditions. The success rate $\mathcal{R}$ is calculated as:

$$\mathcal{R} = \frac{1}{N} \sum_{i=1}^{N} \mathbb{I}\left(\frac{\sum_{j=1}^{M} C_j}{M} > 60\%\right), j = 1, ..., M, \tag{4}$$

where $C_j$ is a binary variable that takes the value of 0 or 1, indicating whether the $j^{th}$ work condition results in non-convergence (0) or convergence (1), and $N$ represents the number of generated airfoils. Here, $\mathbb{I}(x)$ is the indicator function, where $\mathbb{I}(\text{True}) = 1$ and $\mathbb{I}(\text{False}) = 0$.

Table 1: Comparative Performance of our Baselines on Controllable Airfoil Generation Tasks across Different Datasets

| Method | Dataset | Label error↓ ×0.01 | | | | | | | | | | | | $\mathcal{D}\uparrow$ | $\mathcal{M}\downarrow$ ×0.01 |
|---|---|---|---|---|---|---|---|---|---|---|---|---|---|---|---|
| | | $\sigma_1$ | $\sigma_2$ | $\sigma_3$ | $\sigma_4$ | $\sigma_5$ | $\sigma_6$ | $\sigma_7$ | $\sigma_8$ | $\sigma_9$ | $\sigma_{10}$ | $\sigma_{11}$ | $\bar{\sigma}$ | | |
| CVAE [16] | AF-200K | 7.29 | 5.25 | 3.52 | 1590 | 9.9 | 9.55 | 2900 | 1.91 | 1.53 | 4.6 | 10.4 | 413.1 | -155.4 | 7.09 |
| CGAN [15] | AF-200K | 10.7 | 8.50 | 5.44 | 2320 | 14.3 | 13.7 | 5960 | 2.53 | 2.23 | 5.3 | 12.9 | 759.6 | -120.5 | 7.31 |
| PK-VAE | AF-200K | 6.30 | 4.79 | 3.13 | 862 | 6.6 | 6.41 | 1710 | 1.35 | 0.93 | 3.3 | 7.8 | 237.5 | -150.1 | 5.93 |
| PK-GAN | AF-200K | 8.18 | 6.30 | 4.70 | 2103 | 12.0 | 11.7 | 3247 | 2.25 | 1.96 | 5.0 | 12.7 | 492.3 | -112.3 | 3.98 |
| PKVAE-GAN | AF-200K | 5.68 | 3.17 | 3.10 | 565 | 4.6 | 4.35 | 1200 | 0.91 | 0.51 | 2.8 | 6.3 | 163.3 | -129.6 | 2.89 |
| PK-DIFF | AF-200K | 4.61 | 3.46 | 2.15 | 277 | 2.2 | 1.93 | 1030 | 0.70 | 0.11 | 2.4 | 3.1 | 120.6 | -101.3 | 1.52 |
| PK-DIT | UIUC | 6.38 | 5.14 | 3.36 | 1183 | 8.7 | 8.49 | 2570 | 1.69 | 1.19 | 3.6 | 9.8 | 345.6 | -141.7 | 6.03 |
| PK-DIT | Super | 5.20 | 3.50 | 2.40 | 301 | 2.9 | 3.32 | 1050 | 0.83 | 0.26 | 2.7 | 3.3 | 125.0 | -123.4 | 1.97 |
| PK-DIT | AF-200K | **1.12** | **3.23** | **1.54** | **105** | **1.3** | **1.15** | **979** | **0.05** | **0.05** | **2.3** | **2.4** | **99.7** | **-93.2** | **1.04** |

Table 2: Comparison of PK-VAE and PK-VAE$^2$ Performance on Keypoint Editing (EK) and Physical Parameter Editing (EP) Tasks

| Method | Task | Label error↓ ×0.01 | | | | | | | | | | | | $\mathcal{D}\uparrow$ | $\mathcal{M}\downarrow$ ×0.01 |
|---|---|---|---|---|---|---|---|---|---|---|---|---|---|---|---|
| | | $\sigma_1$ | $\sigma_2$ | $\sigma_3$ | $\sigma_4$ | $\sigma_5$ | $\sigma_6$ | $\sigma_7$ | $\sigma_8$ | $\sigma_9$ | $\sigma_{10}$ | $\sigma_{11}$ | $\bar{\sigma}$ | | |
| PK-VAE | EK | 9.3 | 8.33 | 5.27 | 2082 | 12.9 | 11.1 | 4620 | 2.51 | 2.04 | 5.1 | 11.8 | 615.5 | -143.4 | 7.21 |
| PK-VAE | EP | 8.9 | 6.38 | 4.94 | 1780 | 10.9 | 9.4 | 4570 | 2.05 | 1.98 | 4.9 | 10.3 | 582.6 | -150.8 | 7.19 |
| PK-VAE$^2$ | EK | 7.1 | 5.71 | 4.05 | 1430 | 8.0 | 8.1 | 3780 | 1.91 | 1.52 | 3.6 | 8.7 | 478.1 | **-133.4** | **6.20** |
| PK-VAE$^2$ | EP | **6.5** | **5.22** | **3.57** | **1010** | **7.8** | **7.3** | **2010** | **1.52** | **1.03** | **3.4** | **7.9** | **278.5** | -135.6 | 6.36 |

## 5 Benchmarking Results

The baselines in Sec. 4.3 are trained with 500 epochs and a batch size of 512. In the following, we will present the results of our proposed method on controllable airfoil generation and controllable airfoil design, as well as the ablation study to validate the effectiveness of the dataset and methodology.

### 5.1 Comprehensive Method Comparison

**Controllable Airfoil Generation.** We evaluated all our baselines and illustrate the experimental results in Tab. 1, from which we can make the following observations. First, our proposed AF-200K dataset is effective than previous datasets. As the dataset size increased from UIUC 1,600 to Supercritical Airfoil 21,500, and finally to AF-200K, the results indicate that Label Error decreased, Diversity Score increased (indicating generating more diverse airfoils), and Smoothness value decreased (indicating better geometric quality). These results demonstrate that as the size and diversity of the dataset increase, the model performance increases, which validates the effectiveness of our proposed methods. Second, PK-VAE and PK-GAN, PK-VAE exhibits lower Label Error and generates more consistent airfoil shapes, although with reduced diversity due to the strong constraints imposed by the reconstruction loss in VAE. PK-GAN, compared to cGAN, shows by using Bézier curves as intermediate representations, it generates smoother airfoil shapes. PK-VAE-GAN combines the stability of VAE and the diversity of GAN, positioning its performance in between. The Diffusion architecture is simpler and more stable in training compared to VAE and GAN. Comparing PK-DIFF, based on raw data and Unet architecture, with PK-DIT, based on latent space, PK-DIT generates more diverse and smoother airfoil shapes.

**Controllable Airfoil Editing.** For training the airfoil editing task, we randomly sample two airfoils as the source and target. The airfoil editing task is divided into two parts: editing the control points and editing the physical parameters. For editing the control keypoints, the model takes as input (source-physical, target-keypoint) and is expected to output an airfoil that satisfies (target-physical, target-keypoint). For editing the physical properties, the model takes as input (target-physical, source-keypoint) and is expected to output an airfoil that satisfies (target-physical, target-keypoint). The results for these two editing tasks are presented in Table 2. It can be observed that PK-VAE$^2$ outperforms PK-VAE across the board. Specifically, PK-VAE$^2$ achieves a lower label error in physical parameter editing and demonstrates a higher diversity score and better smoothness in keypoint editing.

### 5.2 Ablation Study

**Pretrain and Finetune.** To verify whether the AF-200K dataset can help the model generate airfoils with better aerodynamic capabilities, we select about 20,000 airfoils with superior aerodynamic performance from the AF-200K dataset and then pre-train on the full AF-200K dataset and fine-tune

Table 3: Performance of PK-DIT using data by different generative methods

| Method | Label Error ($\times 0.01$) $\downarrow$ | | | | | | | | | | | | $\mathcal{D} \uparrow$ | $\mathcal{M} \downarrow \times 0.01$ |
| | $\sigma_1$ | $\sigma_2$ | $\sigma_3$ | $\sigma_4$ | $\sigma_5$ | $\sigma_6$ | $\sigma_7$ | $\sigma_8$ | $\sigma_9$ | $\sigma_{10}$ | $\sigma_{11}$ | $\bar{\sigma}$ | | |
|---|---|---|---|---|---|---|---|---|---|---|---|---|---|---|
| NACA-GEN | 6.26 | 5.10 | 3.29 | 961 | 7.69 | 7.46 | 2130 | 1.08 | 1.038 | 3.4 | 8.0 | 284.9 | -136.4 | 5.09 |
| CST-GEN | 5.82 | 4.09 | 2.80 | 572 | 4.61 | 4.36 | 1390 | 0.94 | 0.542 | 3.1 | 5.9 | 181.3 | **-101.5** | 2.31 |
| BézierGAN-GEN | 5.96 | 4.96 | 3.07 | 839 | 5.64 | 6.38 | 1900 | 0.98 | 0.817 | 3.1 | 7.4 | 252.5 | -125.3 | **1.21** |
| Diffusion-GEN | **5.44** | **3.83** | **2.58** | **353** | **3.09** | **3.33** | **1180** | **0.89** | **0.293** | **2.9** | **4.2** | **141.8** | -111.9 | 2.05 |

on this subset. The experimental results illustrate that finetuning on airfoils with high aerodynamic performance can improve the model's success rate from 33.6% to 42.99% (Tab. 4).

**Different Generative Data.** To evaluate the impact of different generative data on the final model performance, we select 10,000 airfoils each from NACA-GEN, CST-GEN, BézierGAN-GEN, and Diffusion-GEN, and train the model on these datasets. The results are shown in Tab. 3. We find that, CST-GEN provided the model with the most diversity. BézierGAN-GEN granted the model the highest score of smoothness. Diffusion-GEN impart the model with the greatest control capability and the lowest label error.

Table 4: Success rate under different training strategies using PK-DIT.

| Strategy | Data | $\mathcal{R}$ |
|---|---|---|
| From Scratch | 160k | 33.60% |
| Finetune | 20k | 42.99% |

## 6 Conclusion

We have proposed a large-scale and diverse airfoil dataset, AF-200K, which has been demonstrated to significantly improve the capabilities of data-driven models compared to previous datasets. Additionally, we have introduced a comprehensive benchmark that evaluates the performance of mainstream generative models on the task of airfoil inverse design. This benchmark provides researchers with a valuable tool to explore more powerful inverse design methods.

As the availability of data continues to expand and AI techniques advance, there is great potential to explore an even broader design space. AI-driven exploration can transcend the limitations of human experience and create innovative structures that are beyond human imagination. In complex design scenarios, AI may achieve superior outcomes compared to human experts. We believe that our methods can also offer valuable insights for 3D airfoil design.

Looking ahead, we aim to establish a more comprehensive benchmark for both 2D and 3D airfoil inverse design. The limitations of our current approach are discussed in Appendix F.

## 7 Acknowledgments

This work is partially supported by Shanghai Artificial Intelligence Laboratory. This work is partially supported by the Natural Science Foundation of China (No. U23A2069).

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

# Appendix

## A  AF-200K Dataset

We publish the AF-200K dataset, benchmark, demo and codebase at our website Page-AFBench. It is our priority to protect the privacy of third parties. We bear all responsibility in case of violation of rights, etc., and confirmation of the data license.

**Terms of use, privacy and License.** The AF-200K dataset is published under the CC BY-NC-SA 4.0, which means everyone can use this dataset for non-commercial research purpose. The original UIUC dataset is released under the GPL license. The original NACA dataset is released under the MIT license.

**Data maintenance.** Data is stored in Google Drive for global users, and the AF-200K is stored in here. We will maintain the data for a long time and check the data accessibility on a regular basis.

**Benchmark and code.** AFBench provides benchmark results and codebase of AF-200K.

**Data statistics.** For AF-200K, there are 160K airfoils for training, 20K airfoils for valuation, 20K airfoils for testing, and 200K airfoils in total.

**Limitations.** The current aerodynamic labels are computed using the relatively coarse CFD solver XFoil. In future work, higher precision CFD simulation software can be utilized to improve the accuracy of the aerodynamic labels.

## B  Background and Related Work

In this section, we primarily discuss three concepts: airfoil design, airfoil representation and conditional generative methods in airfoil design.

### B.1  Airfoil Design

The essence of airfoil design is to find an airfoil that satisfies one's requirements within a vast design space. However, the traditional trial-and-error process is inefficient and costly. To address this issue, a significant amount of research has been conducted, which can be broadly categorized into airfoil parameterization [43], airfoil aerodynamic performance prediction [44, 45], airfoil inverse design [7], and airfoil shape optimization [46]. Airfoil parameterization compresses the airfoil into a few parameters, effectively reducing the design space to a parameter space, which can be searched more quickly. However, parameterization may introduce discontinuities in the design space, making it challenging to find the desired airfoil. Airfoil aerodynamic performance prediction can be divided into two main approaches: using PINNs [47] to solve for the aerodynamic coefficients on the airfoil surface, and employing data-driven surrogate models to quickly predict the performance of the current airfoil, approximating the traditional CFD approach. Airfoil inverse design takes the desired requirements as input and outputs an airfoil that satisfies those requirements. Airfoil shape optimization aims to find the design variables that maximize the lift-to-drag ratio (Cl/Cd).

The ultimate goal of these four directions is to use algorithms to automatically find airfoils that meet the given requirements. Airfoil parameterization can reduce the search variables, airfoil inverse design can provide multiple candidate airfoils as initial values, airfoil aerodynamic performance prediction can use surrogate models for rapid feedback, and airfoil shape optimization can employ optimization methods to find the optimal airfoil. Previous efforts [26, 27] have explored datasets for investigating airfoil aerodynamic characteristics, but have largely relied on the UIUC and NACA airfoil shapes, lacking the breadth of data necessary to support the exploration of large-scale airfoil models. Our dataset, on the other hand, boasts a more diverse collection of airfoils and rich annotations, enabling better support for the four research directions mentioned earlier. Additionally, we have proposed AFBench, which includes airfoil generation and airfoil editing. Airfoil generation is a combination and complement to inverse design and parameterization, as it can generate airfoils that satisfy geometric constraints based on PARSEC parameters. Airfoil editing is a supplement to shape optimization, as it allows designers to more controllably find the optimal airfoil based on their experience.

## B.2 Airfoil Representation

Airfoil representation is an evolution of airfoil parameterization. It can be broadly categorized into explicit representation and implicit representation. Explicit representation includes the most common Coordinate Point Method, as well as polynomial-based Parametric Representation Methods, such as PARSEC [37], Bézier [35], CST [32]. The former is the easiest to manipulate, but the large number of variables makes it difficult to optimize. The polynomial-based representations can reduce the design variables while ensuring the represented airfoils are smooth. Some, like PARSEC, even have intuitive geometric interpretations, such as leading edge radius and upper/lower surface peak values. Other parameterization methods, however, have design variables that are less intuitive. Additionally, their design spaces tend to be relatively small.

Implicit representation primarily uses data-driven methods to compress the airfoil into a latent space. Traditional methods include SVD [48] and PCA [49], but these linear combination approaches also result in small design spaces. More recently, neural representations have become common, where a well-trained neural network can store the airfoil information, allowing the design space to be sampled from a low-dimensional space. A representative work in this area is BézierGAN [33]. Our work adopts a hybrid approach, combining implicit and explicit representations. By adjusting the intuitive PARSEC parameters and control points, we can achieve airfoil generation. The neural network representation allows for a much larger design space compared to pure PARSEC parameterization.

## B.3 Conditional Generative methods in Airfoil Design

Leveraging the advantages of both implicit representation and generative methods, there recently appears attempts to combine implicit representation and generative methods to achieve better design performance.

**VAE** Variation Auto Encoder[28] trains a model to minimize reconstruction loss and latent loss, and it is usually optimized considering the sum of these losses. [22] proposes two advanced CVAE for the inverse airfoil design problems by combining the conditional variational autoencoders (CVAE) and distributions. There are two versions: N-CVAE, which combines the CVAE with normal distribution [50], and S-CVAE, which combines the VAE and von Mises-Fischer distribution [51]. Both the CVAE models convert the original airfoils into a latent space. Differently, the S-CVAE enables the separation of data in the latent space, while the N-CVAE embeds the data in a narrow space. These different features are used for various tasks.

**GAN** Generative adversarial network [29] uses a generative neural network to generate a airfoil and uses a discriminative neural network to justify the airfoil is real or fake. CGAN [15] improves the original GAN by inputing the conditions to both the generator and discriminator. Then, the generator is learned to generate the shape satisfying the condition constraints. Inspired by the big success in computer vision, several works are proposed to explore the applications of GANs to solve the airfoil design. [30] generates shapes with low or high lift coefficient. By inputing the aerodynamic characteristics such as lift-to-drag ratio (Cl/Cd) or share parameters, it is possible to guide the shape generation process toward particular airfoil.

**Diffusion** Diffusion model [17] is the recent widely adapted generative model in the image and 3D computer vision. In the airfoil generation field, there are also several initial attempts to use diffusion model to generate airfoils. [14] uses conditional diffusion models to perform performance-aware and manufacturability-aware topology optimization. Specifically, a surrogate model-based guidance strategy is proposed that actively favors structures with low compliance and good manufacturability. [13] introduces compositional inverse design with diffusion models, which enables the proposed method to generalize to out-of-distribution and more complex design inputs than seen in training. [31] leverages the capable of a latent denoising diffusion model to generate airfoil geometries conditioned on flow parameters and an area constraint. They found that the diffusion model achieves better generation performance than GAN-based methods.

# C  Detailed Description of Airfoil Generation by CST Method

The CST-assisted Generation Stage augments the airfoils with physical models, *i.e.,* CST model. Given one airfoil $f_0$, we generate the airfoils with assistance of CST by the following steps: (1) We parameterize the airfoil $f_0$ with the CST method [32] as $p_0$; (2) Employ Latin Hypercube Sampling

(LHS) to perturb the CST parameters. LHS enables uniform sampling across the CST parameter space, thereby facilitating the generation of new airfoils.

**Parametrize the airfoils using CST** Given one manually designed airfoil $f_0$, we parameterize the airfoil with the CST method as $p_0 = (p_0^1, p_0^2, ..., p_0^M)$, where $M$ is the number of physical parameters. The CST method is a widely-accepted method in airfoil design and proposes to fit the supercritical airfoil with the Bernstein polynomial, which can be mathematically expressed as

$$\zeta(\psi) = C(\psi)S(\psi) + \psi\zeta_T, \tag{5}$$

where $\zeta = Y/c$, $\psi = X/c$, and $c$ represents the chord length. $\zeta_T = \Delta\zeta_{TE}/c$ represents the trailing edge thickness of the airfoil. Here, $X$ and $Y$ are x-corrdinates and y-coordinates of the airfoil. $C(\psi)$ and $S(\psi)$ correspond to the class function and shape function, respectively, which can be formally described as follows:

$$C(\psi) = (\psi)^{N_1}(1 - \psi)^{N_2} \tag{6}$$

$$S(\psi) = \frac{\zeta(\psi) - \psi\zeta_T}{\sqrt{\psi(1 - \psi)}} \tag{7}$$

where $N_1$ and $N_2$ define the class of airfoils. In this paper, we choose $N_1 = 0.5$ and $N_2 = 1.0$ to represent the circular leading edge and sharp trailing edge of supercritical airfoils.

**Perturb the CST parameters using Latin Hypercube Sampling (LHS) and generate new airfoils** Given the parameterized supercritical airfoil, we perturb the parameters of the airfoils with Latin hypercube sampling (LHS). Take generating $N$ airfoils based on one manually-designed airfoil for example. For every variable in our parameterized airfoil, we evenly divided into $N$ parts, and randomly sample a value in $N$ parts, respectively for $N$ generated airfoils. With Latin hypercube sampling (LHS), the generated airfoils can be uniformly sampled from the parametric space of CST for supercritical airfoils. The generated aifoils are illustrated in Fig. 3.

# D    Detailed Description of Airfoil Generation by Generative Models

**BézierGAN-GEN** BézierGAN [33] uses a Bézier layer to transform the network's predicted control points, weights, and parameter variables into smooth airfoil coordinates. The Bézier layer formula is as follows:

$$X_j = \frac{\sum_{i=0}^n \binom{n}{i} u_j^i (1 - u_j)^{n-i} P_i w_i}{\sum_{i=0}^n \binom{n}{i} u_j^i (1 - u_j)^{n-i} w_i}, \quad j = 0, \ldots, m \tag{8}$$

where $n$ is the Bézier degree, $m + 1$ is the number of airfoil coordinate points, and $P_i, w_i, u_j$ are the network-predicted control points, weights, and parameter variables, respectively, which are all differentiable.

By applying the aforementioned models, we uniformly sampled 10,000 latent codes from the range [0, 1], combined them with Gaussian noise to form the input z, and used the generator to produce 10,000 smooth airfoil shapes represented as 257 x 2 coordinate points.

**Diff-GEN** Diffusion models [17] are a recent advancement in a generative modeling. Compared to the traditional GAN models, diffusion models introduce the following main improvements: For Noise Schedule: Instead of learning to transform noise to data directly, diffusion models gradually transform noise into data through a series of small, reversible steps, which makes the training more stable and produces higher-quality samples. For the model architecture, the generator is replaced with a U-Net architecture that predicts the noise added to the data at each step, while a diffusion process progressively refines this prediction. The forward diffusion process formula is as follows:

$$q(x_t|x_{t-1}) = \mathcal{N}(x_t; \sqrt{\alpha_t}x_{t-1}, (1 - \alpha_t)I), t = 1, 2, \cdots, T \tag{9}$$

where $T$ is the total number of diffusion steps, $\alpha_t$ is a variance schedule controlling the amount of noise added at each step, and $\mathcal{N}$ denotes a normal distribution. The model learns to reverse this process, thereby generating samples from pure noise through a series of learned denoising steps.

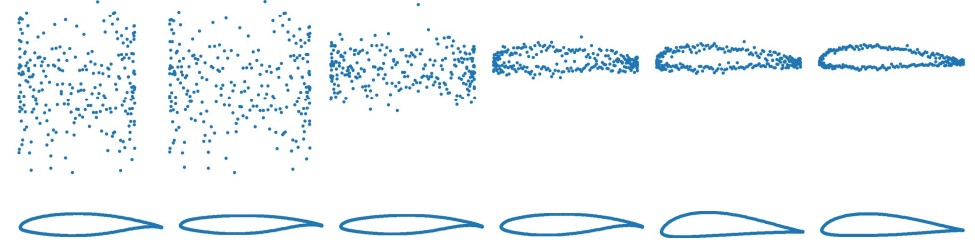

Figure 7: Reverse diffusion process of raw data and latent data. The top row shows the visualization of PK-DIFF samples, while the bottom row shows the visualization of PK-DIT samples.

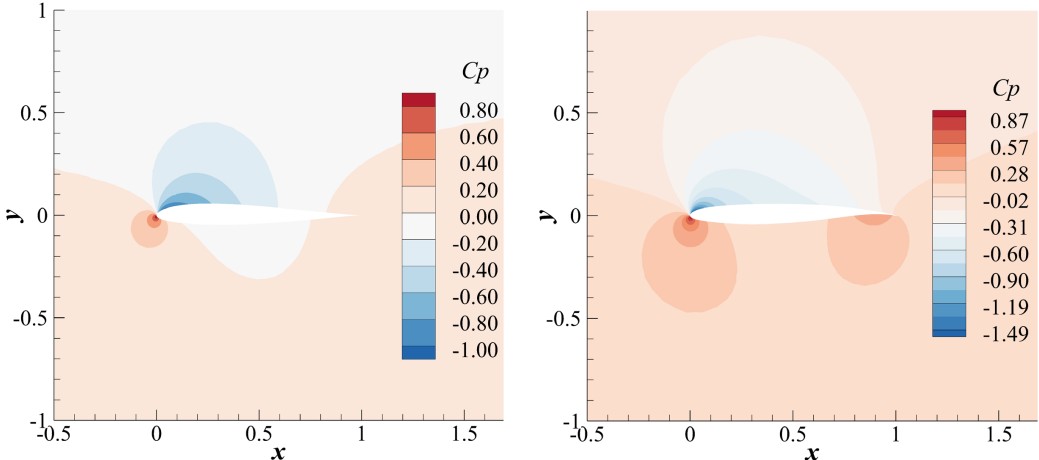

Figure 8: Aerodynamic Performance Visualization between PK-DIFF (left) and PK-DIT (right).

Specifically for airfoil generation, we need to first encode the 257 x 2 airfoil coordinates into a 32x1 latent variable using a pretrained VAE. The diffusion model then learns to generate these latent variables, which are subsequently decoded by the VAE to produce the final airfoil shape. This approach leverages the strengths of both VAEs and diffusion models, where the VAE efficiently compresses the high-dimensional airfoil data into a more manageable latent space, and the diffusion model excels at generating high-quality samples within this latent space. By combining these methods, we achieve a robust and efficient framework for generating realistic and high-quality airfoil designs.

## E    Diffusion Results

**Reverse Diffusion Process.** Fig. 7 illustrates the denoising sampling results of the Diffusion model at different time steps. It can be observed that when trained directly on raw data, the generated airfoils are not smooth. In contrast, airfoils trained in the latent space are smooth from the beginning due to the pre-trained VAE providing a performance baseline. As the reverse steps increase, the generated airfoils gradually align more closely with the given physical conditions.

**Aerodynamic Performance Visualization.** Given the same conditions, airfoils were generated using both PK-DIFF and PK-DIT. We used a refined CFD solver OpenFOAM to calculate the flow and aerodynamic performance of these two generated airfoils. Fig. 8 shows the distribution of the pressure coefficient around the airfoils generated by PK-DIFF and PK-DIT. Under the given working conditions [AoA = 3°, Re = 1e6], the lift coefficient (Cl) and drag coefficient (Cd) for PK-DIFF are (0.36, 0.01029), while for PK-DIT, they are (0.7335, 0.0125). The higher Cl/Cd ratio for PK-DIT indicates that the airfoil generated by PK-DIT has superior aerodynamic performance.

**Generate Diverse Airfoils by PK-DIT** Fig. 9 illustrates the diversity of airfoils generated by the Diffusion model. Starting from random noise, Diffusion progressively denoises the airfoil. Each

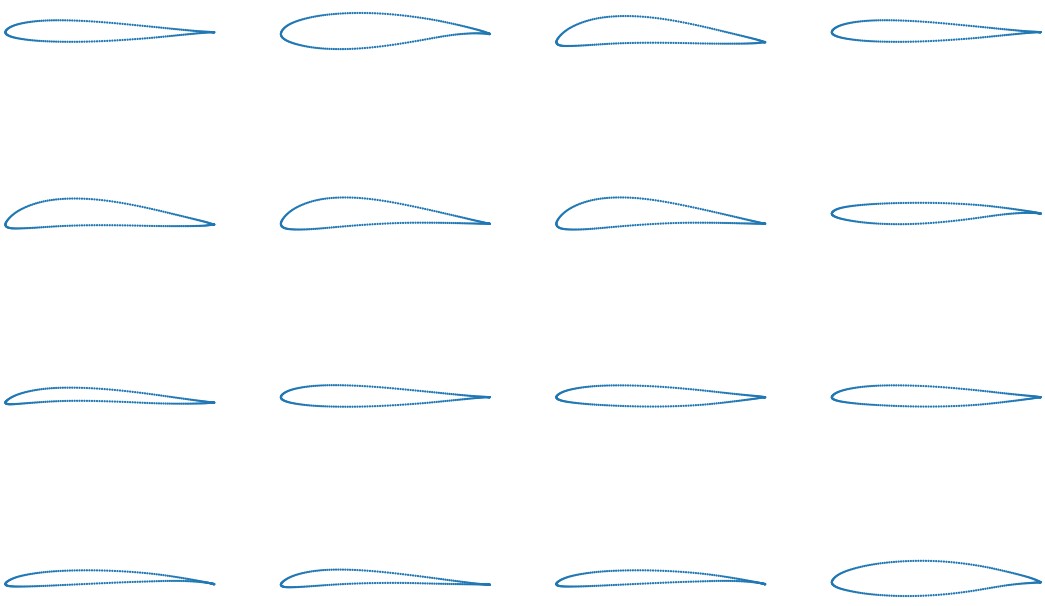

Figure 9: use PK-DIT generated diverse novel airfoil.

denoising step can introduce different details, thus ensuring that the generated airfoils are diverse while still meeting the required conditions.

## F   Limitation

The current physical parameters and control keypoints used in our approach are coupled within each condition. For example, simply changing leading edge radius may not result in a feasible airfoil, as the design space may not contain such a configuration. When dealing with multiple conditions, finding and balancing the conflicts between the conditions to generate an optimal airfoil is a challenge that remains unsolved and deserves further exploration. In addition to finding better airfoil design variables, modeling the relationships between these variables is also crucial. Moreover, our method currently does not integrate airfoil shape optimization techniques into the generation process. Embedding optimization methods to produce generated airfoils with superior aerodynamic performance, surpassing manually designed airfoils, would further demonstrate the effectiveness of AI-based approaches, and is another area worth investigating.

## G   Datasheet

1. **For what purpose was the dataset created?** Was there a specific task in mind? Was there a specific gap that needed to be filled? Please provide a description.

   - AFBench was created as a benchmark for airfoil inverse design task. The goal of this task is to find the design input variables that optimize a given objective function. Although some related datasets and works have been proposed, they do not take into account the real needs of applications. Moreover, there is still a lack of large-scale foundational data and evaluation metrics.

2. **Who created the dataset (e.g., which team, research group) and on behalf of which entity (e.g., company, institution, organization)?**

- This dataset is presented by HIT-AIIA Lab & Shanghai AI Lab & Shanghai Aircraft Design and Research Institute.

3. **Who funded the creation of the dataset?** If there is an associated grant, please provide the name of the grantor and the grant name and number.

   - This work was sponsored by Shanghai AI Lab.

4. **Any other comments?**

   - No.

### G.1 Composition

5. **What do the instances that comprise the dataset represent (e.g., documents, photos, people, countries)?** *Are there multiple types of instances (e.g., movies, users, and ratings; people and interactions between them; nodes and edges)? Please provide a description.*

   - AFBench comprises existing UIUC and NACA datasets, along with 2,150 manually designed supercritical airfoils and airfoils generated by models, totaling 200k samples. Each sample consists of a well-designed airfoil, accompanied by 11 geometric parameters and aerodynamic properties under 66 work conditions. We made our benchmark openly available on the AFBench github page(https://hitcslj.github.io/afbench/).

6. **How many instances are there in total (of each type, if appropriate)?**

   - For AF-200K, there are 160K airfoils for training, 20K airfoils for valuation, 20K airfoils for testing, 200K airfoils in total.

7. **Does the dataset contain all possible instances or is it a sample (not necessarily random) of instances from a larger set?** *If the dataset is a sample, then what is the larger set? Is the sample representative of the larger set (e.g., geographic coverage)? If so, please describe how this representativeness was validated/verified. If it is not representative of the larger set, please describe why not (e.g., to cover a more diverse range of instances, because instances were withheld or unavailable).*

   - Both UIUC and NACA are open-source datasets. We use the proposed CST method and unconditional generative models to derive AF-200K dataset. For AF-200K, we use all samples of UIUC and NACA Open dataset.

8. **What data does each instance consist of?** *"Raw" data (e.g., unprocessed text or images) or features? In either case, please provide a description.*

   - Each instance consists of a well-designed airfoil, accompanied by 11 geometric parameters and aerodynamic properties under 66 work conditions.

9. **Is there a label or target associated with each instance?** *If so, please provide a description.*

   - Each instance includes various aerodynamic performance metrics such as angle of attack (AoA), drag coefficient (CD), and moment coefficient (CM), under different work conditions. Additionally, PARSEC physical parameters are provided as geometric features for each instance.

10. **Is any information missing from individual instances?** *If so, please provide a description, explaining why this information is missing (e.g., because it was unavailable). This does not include intentionally removed information, but might include, e.g., redacted text.*

    - No.

11. **Are relationships between individual instances made explicit (e.g., users' movie ratings, social network links)?** *If so, please describe how these relationships are made explicit.*

    - No.

12. **Are there recommended data splits (e.g., training, development/validation, testing)?** *If so, please provide a description of these splits, explaining the rationale behind them.*

    - We recommend using the default 8:1:1 ratio provided by AFBench for dataset partitioning.

13. **Are there any errors, sources of noise, or redundancies in the dataset?** *If so, please provide a description.*

    - No.

14. **Is the dataset self-contained, or does it link to or otherwise rely on external resources (e.g., websites, tweets, other datasets)?** *If it links to or relies on external resources, a) are there guarantees that they will exist, and remain constant, over time; b) are there official archival versions of the complete dataset (i.e., including the external resources as they existed at the time the dataset was created); c) are there any restrictions (e.g., licenses, fees) associated with any of the external resources that might apply to a future user? Please provide descriptions of all external resources and any restrictions associated with them, as well as links or other access points, as appropriate.*

    - We release the AFBench dataset on our GitHub repository: `https://github.com/hitcslj/AFBench`. More specifically, please follow the instructions provided on the website: AFBench-Webpage. Our dataset is developed based on existing airfoil dataset UIUC and NACA.

15. **Does the dataset contain data that might be considered confidential (e.g., data that is protected by legal privilege or by doctor–patient confidentiality, data that includes the content of individuals' non-public communications)?** *If so, please provide a description.*

    - No.

16. **Does the dataset contain data that, if viewed directly, might be offensive, insulting, threatening, or might otherwise cause anxiety?** *If so, please describe why.*

    - No.

17. **Does the dataset relate to people?** *If not, you may skip the remaining questions in this section.*

    - No.

18. **Does the dataset identify any subpopulations (e.g., by age, gender)?**

    - No.

19. **Is it possible to identify individuals (i.e., one or more natural persons), either directly or indirectly (i.e., in combination with other data) from the dataset?** *If so, please describe how.*

    - No.

20. **Does the dataset contain data that might be considered sensitive in any way (e.g., data that reveals racial or ethnic origins, sexual orientations, religious beliefs, political opinions or union memberships, or locations; financial or health data; biometric or genetic data; forms of government identification, such as social security numbers; criminal history)?** *If so, please provide a description.*

    - No.

21. **Any other comments?**

    - No.

### G.2    Collection Process

22. **How was the data associated with each instance acquired?** *Was the data directly observable (e.g., raw text, movie ratings), reported by subjects (e.g., survey responses), or indirectly inferred/derived from other data (e.g., part-of-speech tags, model-based guesses for age or language)? If data was reported by subjects or indirectly inferred/derived from other data, was the data validated/verified? If so, please describe how.*

    - Our data is developing based on published airfoil dataset UIUC and NACA using a designed CST method and unconditional generative models mentioned before.

23. **What mechanisms or procedures were used to collect the data (e.g., hardware apparatus or sensor, manual human curation, software program, software API)?** *How were these mechanisms or procedures validated?*

    - For UIUC, we write a Python script that uses Bézier interpolation to generate a smooth airfoil with a specified number of points. For NACA, we write an NACA generator script to sample the airfoil at a specified number of points. For the rest, we use CST and generative methods to generate the airfoils, then use XFoil to create the aerodynamic labels, and a Python script to calculate the geometry labels. We use hundreds of small CPU nodes and small GPU nodes for the computation.

24. **If the dataset is a sample from a larger set, what was the sampling strategy (e.g., deterministic, probabilistic with specific sampling probabilities)?**

    - We use full-set provided by UIUC and NACA.

25. **Who was involved in the data collection process (e.g., students, crowdworkers, contractors) and how were they compensated (e.g., how much were crowdworkers paid)?**

    - No crowdworkers were involved in the curation of the dataset. Open-source researchers and developers enabled its creation for no payment.

26. **Over what timeframe was the data collected? Does this timeframe match the creation timeframe of the data associated with the instances (e.g., recent crawl of old news articles)?** *If not, please describe the timeframe in which the data associated with the instances was created.*

    - The AF-200K data and label was generated in 2024, while the source data UIUC v2 was created in 2020, NACA v1 was created in 1933.

27. **Were any ethical review processes conducted (e.g., by an institutional review board)?** *If so, please provide a description of these review processes, including the outcomes, as well as a link or other access point to any supporting documentation.*

    - The source sensor data for UIUC and NACA had been conducted ethical review processes by UIUC Applied Aerodynamics Group and National Advisory Committee for Aeronautics airfoils, which can be referred to UIUC and NACA, respectively.

28. **Did you collect the data from the individuals in question directly, or obtain it via third parties or other sources (e.g., websites)?**

    - We retrieve the data from the open source dataset UIUC and NACA.

29. **Were the individuals in question notified about the data collection?** *If so, please describe (or show with screenshots or other information) how notice was provided, and provide a link or other access point to, or otherwise reproduce, the exact language of the notification itself.*

    - The AFBench dataset is developed based on open-source dataset and following the open-source license.

30. **Did the individuals in question consent to the collection and use of their data?** *If so, please describe (or show with screenshots or other information) how consent was requested and provided, and provide a link or other access point to, or otherwise reproduce, the exact language to which the individuals consented.*

    - The AFBench dataset is developed on open-source dataset and obey the license.

31. **If consent was obtained, were the consenting individuals provided with a mechanism to revoke their consent in the future or for certain uses?** *If so, please provide a description, as well as a link or other access point to the mechanism (if appropriate).*

    - Users have a possibility to check for the presence of the links in our dataset leading to their data on public internet by using the search tool provided by AFBench, accessible at AFBench-Webpage. If users wish to revoke their consent after finding sensitive data, they can contact the hosting party and request to delete the content from the underlying website. Please leave the message in GitHub Issue to request removal of the links from the dataset.

32. **Has an analysis of the potential impact of the dataset and its use on data subjects (e.g., a data protection impact analysis) been conducted?** *If so, please provide a description of this analysis, including the outcomes, as well as a link or other access point to any supporting documentation.*

    • We develop our dataset based on open source dataset UIUC and NACA publised by UIUC Applied Aerodynamics Group and National Advisory Committee for Aeronautics airfoils. The published dataset has been seriously considered of it's potential impact and its use on data subjects.

33. **Any other comments?**

    • No.

### G.3 Preprocessing, Cleaning, and/or Labeling

34. **Was any preprocessing/cleaning/labeling of the data done (e.g., discretization or bucketing, tokenization, part-of-speech tagging, SIFT feature extraction, removal of instances, processing of missing values)?** *If so, please provide a description. If not, you may skip the remainder of the questions in this section.*

    • Above all, We utilize B-spline interpolation to convert the discrete points into a continuous representation. Then we use CST method to augment the dataset and XFOIL to calculate the corresponding aerodynamic labels. Additionally, We utilize PARSEC physical parameters with Control keypointsBeside as Geometric label. Besides this, no preprocessing or labelling is done.

35. **Was the "raw" data saved in addition to the preprocessed/cleaned/labeled data (e.g., to support unanticipated future uses)?** *If so, please provide a link or other access point to the "raw" data.*

    • Yes, we provide the original open source dataset and the augmented AF-200K dataset.

36. **Is the software used to preprocess/clean/label the instances available?** *If so, please provide a link or other access point.*

    • Yes, XFOIL is accessible at https://github.com/hitcslj/Xfoil-cal.

37. **Any other comments?**

    • No.

### G.4 Uses

38. **Has the dataset been used for any tasks already?** *If so, please provide a description.*

    • No.

39. **Is there a repository that links to any or all papers or systems that use the dataset?** *If so, please provide a link or other access point.*

    • No.

40. **What (other) tasks could the dataset be used for?**

    • We encourage researchers to explore more diverse airfoil generation and editing, as well as optimization design.

41. **Is there anything about the composition of the dataset or the way it was collected and preprocessed/cleaned/labeled that might impact future uses?** *For example, is there anything that a future user might need to know to avoid uses that could result in unfair treatment of individuals or groups (e.g., stereotyping, quality of service issues) or other undesirable harms (e.g., financial harms, legal risks) If so, please provide a description. Is there anything a future user could do to mitigate these undesirable harms?*

    • No.

42. **Are there tasks for which the dataset should not be used?** *If so, please provide a description.*

- Due to the known biases of the dataset, under no circumstance should any models be put into production using the dataset as is. It is neither safe nor responsible. As it stands, the dataset should be solely used for research purposes in its uncurated state.

43. **Any other comments?**

   - No.

### G.5    Distribution

44. **Will the dataset be distributed to third parties outside of the entity (e.g., company, institution, organization) on behalf of which the dataset was created?** *If so, please provide a description.*

   - Yes, the dataset will be open-source.

45. **How will the dataset be distributed (e.g., tarball on website, API, GitHub)?** *Does the dataset have a digital object identifier (DOI)?*

   - The data is available through `https://github.com/hitcslj/AFBench`.

46. **When will the dataset be distributed?**

   - 06/2024 and onward

47. **Will the dataset be distributed under a copyright or other intellectual property (IP) license, and/or under applicable terms of use (ToU)?** *If so, please describe this license and/or ToU, and provide a link or other access point to, or otherwise reproduce, any relevant licensing terms or ToU, as well as any fees associated with these restrictions.*

   - The AFBench dataset is published under CC BY-NC-SA 4.0, which means everyone can use this dataset for non-commercial research purpose. The original UIUC dataset is released under the GPL license. The original NACA dataset is released under the MIT license.

48. **Have any third parties imposed IP-based or other restrictions on the data associated with the instances?** *If so, please describe these restrictions, and provide a link or other access point to, or otherwise reproduce, any relevant licensing terms, as well as any fees associated with these restrictions.*

   - The original UIUC dataset is released under the GPL license, and the for the restrictions, please refer to UIUC. The original NACA dataset is released under the MIT license, and the for the restrictions, please refer to NACA

49. **Do any export controls or other regulatory restrictions apply to the dataset or to individual instances?** *If so, please describe these restrictions, and provide a link or other access point to, or otherwise reproduce, any supporting documentation.*

   - No.

50. **Any other comments?**

   - No.

### G.6    Maintenance

51. **Who will be supporting/hosting/maintaining the dataset?**

   - Shanghai AILab will support hosting of the dataset.

52. **How can the owner/curator/manager of the dataset be contacted (e.g., email address)?**

   - `https://github.com/hitcslj/AFBench/issues`

53. **Is there an erratum?** *If so, please provide a link or other access point.*

   - There is no erratum for our initial release. Errata will be documented as future releases on the dataset website.

54. **Will the dataset be updated (e.g., to correct labeling errors, add new instances, delete instances)?** *If so, please describe how often, by whom, and how updates will be communicated to users (e.g., mailing list, GitHub)?*

- We will continue to support AFBench dataset.

55. **If the dataset relates to people, are there applicable limits on the retention of the data associated with the instances (e.g., were individuals in question told that their data would be retained for a fixed period of time and then deleted)?** *If so, please describe these limits and explain how they will be enforced.*

   - No.

56. **Will older versions of the dataset continue to be supported/hosted/maintained?** *If so, please describe how. If not, please describe how its obsolescence will be communicated to users.*

   - Yes. We will continue to support AFBench dataset in our github page.

57. **If others want to extend/augment/build on/contribute to the dataset, is there a mechanism for them to do so?** *If so, please provide a description. Will these contributions be validated/verified? If so, please describe how. If not, why not? Is there a process for communicating/distributing these contributions to other users? If so, please provide a description.*

   - Yes, they can driectly developing on open scource dataset UIUC and NACA dataset or concat us via GitHub Issue.

58. **Any other comments?**

   - No.

