# OpenReview forum: "AFBench: A Large-scale Benchmark for Airfoil Design"
_NeurIPS.cc/2024/Datasets_and_Benchmarks_Track — NeurIPS 2024 Track Datasets and Benchmarks Poster_

### Official Review · Reviewer_nSXb · 2024-07-24
**AFBench Review**

**Rating:** 8
**Confidence:** 4
**Correctness:** Yes
**Clarity:** Yes

**Review:**

The author introduced the data collection/generation process in detail and considered filtering of poor data to ensure the general quality of the dataset. Also, the author designed some evaluation metrics and tested the generation task on many baseline methods. Overall, the paper is well-organized and contains details related to important factors in airfoil design and generation.

**Strengths:**

See 'Review'

**Additional Feedback:**

A minor question is how many airfoils are from COMAC engineers and how many are generated based on these airfoils? It seems that only the total number is provided in line 185.

**Documentation:**

Yes

**Limitations:**

The author has analyzed the limitations in detail.

**Opportunities For Improvement:**

It seems that in line 149 the increment for the Mach number is 0.1, which becomes clear only when reading the total work conditions are 66 in Section 3.3. I guess the increment 0.2 specified for CL led to this confusion.

**Relation To Prior Work:**

Yes

**Summary And Contributions:**

This paper proposes AFBench, which is an airfoil dataset that includes 200K airfoil shapes with geometric and aerodynamic annotation labels. The dataset consists of airfoils from the UIUC and NACA datasets, COMAC engineer designs, and augmented airfoils generated by CST, GAN, and diffusion models. Benchmarks are performed in the airfoil generation tasks.

---

> ### Author Rebuttal · Authors · 2024-08-17
>
> **Q1:** Confusion of the Mach number increment.
>
> **A1:** Sorry for the confusion. The increment for the Mach number is indeed 0.1, which we will note in the final version.
>
> **Q2:** Specific amount of data from COMAC.
>
> **A2:** We obtained 2,000 hand-designed airfoils from the COMAC and further generated physically meaningful 21,000 airfoils after careful filtering (L169-L176).

---

> ### Author Response · Authors · 2024-08-27
>
> Dear Reviewer nSXb,
>
> Thank you for sharing your valuable comments and insights with us! We are eager to learn if our responses have adequately addressed your concerns. Your further feedback is highly appreciated, as it will help us improve our manuscript. If you have any additional questions or comments, please do not hesitate to let us know.
>
> Best regards,
>
> The Authors

---

### Official Review · Reviewer_c69g · 2024-07-25
**Review of "AFBench: A Large-scale Benchmark for Airfoil Design"**

**Rating:** 5
**Confidence:** 3
**Correctness:** Yes.
**Clarity:** Yes.

**Review:**

**Quality**: The paper is well-documented and methodically detailed.

**Clarity**: Not clear and organized, and some technical jargon may be difficult for non-specialists.

**Originality**: Limited. The novelty is limited since the construction of this dataset employs some common methods.

**Significance**: Offers significant value by providing a comprehensive benchmark for airfoil design research.

**Pros**:
- A large-scale dataset.
- Provides an open-source benchmark codebase.

**Cons**:
- Contains specialized terminology which has not been properly introduced, e.g., CST.
- Dataset has acknowledged biases.
- The codebase is not completely open-sourced. For example, the evaluation code has not been released yet. (I have checked the GitHub repo.)

**Strengths:**

The AFBench dataset is a significant contribution, providing a resource-rich benchmark that greatly aids airfoil inverse design research. Its comprehensive data and robust annotations ensure high research quality.

The specialized research goal hinders this work from the broader research community.

Socially, the work encourages open scientific collaboration by offering an open-source codebase and publicly accessible data under a non-commercial license.

**Additional Feedback:**

Is the quality of the synthetic data in the proposed dataset guaranteed?

**Documentation:**

Yes.

**Ethics:**

There is no such suspect.

**Limitations:**

Yes.

**Opportunities For Improvement:**

**Significance**
While the dataset, AFBench, is extensive, it is partially based on a relatively coarse CFD solver, XFoil, which could affect the precision of aerodynamic labels.

**Relevance**
The dataset contains biases and may not generalize well to all real-world applications, limiting its immediate practical use.

**Quality**
Technical complexities and specialized jargon could hinder broader accessibility and understanding among non-expert researchers.

**Relation To Prior Work:**

Yes.

**Summary And Contributions:**

This paper introduces a comprehensive dataset, AFBench, for airfoil inverse design comprising 200,000 airfoils with detailed geometric parameters and aerodynamic properties. Additionally, the authors provide an open-source codebase for evaluating data-driven generative models in airfoil design. Their contributions include high-quality annotations, automatic data synthesis, and comprehensive evaluation metrics. This work aims to accelerate research and comparison in airfoil inverse design methodologies.

---

> ### Author Rebuttal · Authors · 2024-08-17
>
> **Q9:** whether the quality of the synthetic data in the proposed dataset is guaranteed.
>
> **A9:** The quality of synthetic data is guaranteed by our proposed data filtering pipeline. Specifically, we use a numerical solver to simulate and evaluate the aerodynamic performance of the generated airfoil under different working conditions and filter the airfoil that does not meet the expected requirements (success under various working conditions). The detailed process is described in Section 3.3.

---

> ### Author Rebuttal · Authors · 2024-08-17
>
> **Q8:** Technical complexities and specialized jargon.
>
> **A8:** Thank you for your feedback.
> First, we have provided explainations of some critical aerospace jargons about aerospace technology in our paper, such as Leading Edge Radius, CST, which you can refer to A1 above for details. According to your suggestions, we will also add more detailed explaianations of other jargons in the supplementary materials. Please refer to A1 for detailed information.

---

> ### Author Rebuttal · Authors · 2024-08-17
>
> **Q7:** The dataset contains biases and may not generalize well to all real-world applications.
>
> **A7:** Bias in line 805 indicates that our data set intentionally exclude some war-relataed airfoils, such as airfoils used in fighter aircrafts, bombers, and carrier-based aircrafts. We do not wish that the dataset is used for military use. However, our dataset is diverse and can generalize well to airfoils used in Airliners (for civil aviation purpose). This dataset consists of supercritical airfoils, common arifoils and low-speed airfoils that are manually designed by professional engineers. In addition, we collect public data, and synthesize large-scale data by parameterization and generative models, such as diffusion models, based on the collected supercritical airfoils, common arifoils and low-speed airfoils. Please see A4 for more details.

---

> ### Author Rebuttal · Authors · 2024-08-17
>
> **Q6:** Coarse CFD solver might affect the precision of aerodynamic labels.
>
> **A6:** Thanks for your review.
>
> Although xfoil is relatively coarse CFD solvers, it can largely approximate some aerodynamical properties, e.g., lift-to-drag ratio (CL/CD). Specifically, we randomly select 100 airfoils from the training set, and calculate the flow field with xfoil and advanced close-sourced CFD software  Autodesk CFD 2019. According to some comprehensive benchmarking papers [A, B], the flow field calculated by the nonlinear BET coupled with Xfoil is less than 10% relative error to experimental data.  We also validate that the difference of flow fields generated by nonlinear BET coupled with Xfoil and Autodesk CFD 2019 are about 5% relative error. Therefore, we consider the nonlinear BET coupled with Xfoil used in our paper can generate relatively accurate aerodynamic performance of airfoils that can approximate experimental results.
>
> Furthermore, running high-quality CFD software is extremely time-inefficient. Specifically, given an airfoil and a working condition, we need to spend about 10-20 minutes to get its aerodynamic parameters by commercial software such as Fluent [C].  We are preparing more accurate aerodynamical labels computed by Fluent, and promise to release them after careful checking.
>
> [A] Comparison and evaluation of blade element methods against RANS simulations and test data, CEAS Aeronautical Journal (2022) 13:535–557
>
> [B] To CFD or not to CFD? Comparing RANS and viscous panel methods for airfoil shape optimization, Congress of the International Council of the Aeronautical Sciences (ICAS 2022), September 2022, Stockholm, Sweden.
>
> [C] https://www.ansys.com/products/fluids/ansys-fluent

---

> ### Author Rebuttal · Authors · 2024-08-17
>
> **Q5:** The codebase is not completely open-sourced.
>
> **A5:** Thanks for your review. The evaluate code has been uploaded in the repository (https://github.com/open-sciencelab/Intern-WingWing/blob/main/evaluate.py).

---

> ### Author Rebuttal · Authors · 2024-08-17
>
> **Q4:** Dataset has acknowledged biases.
>
> **A4:** We would like to clarify that "Bias" in line 805 means that our datasets intentionally exclude some war-relataed airfoils, such as airfoils used in fighter aircrafts, bombers, and carrier-based aircrafts. We do not wish that the dataset is used for military use.
> In fact, our dataset is diverse for airfoils used in Airliners (for civil aviation purpose), consisting of public data, synthetic data by both traditional parameteric models and unconditional generative models. These data consists of supercritical airfoils, low-speed airfoils and common airfoils, which can generalize to real-world application for civil aviation purpose.

---

> ### Author Rebuttal · Authors · 2024-08-17
>
> **Q3:** Contains specialized terminology which has not been properly introduced, e.g., CST.
>
> **A3:** As detailed in Appendix C, the CST (Class Shape Transformation) method is an aerodynamic technique used to model and transform airfoil shapes. It employs a set of basic functions to fit and adjust the airfoil shape based on the given coordinates, facilitating the generation of new airfoil shapes while preserving aerodynamic properties. For a specific implementation of CST parameterization, please refer to our code available at CST Implementation Code (https://github.com/open-sciencelab/Intern-WingWing/blob/main/datagen/cst_gen.py).

---

> ### Author Rebuttal · Authors · 2024-08-17
>
> **Q2:** The novelty is limited since the construction of this dataset employs some common methods.
>
> **A2:** Thanks for your review.
>
> We respectively disagree that our novelty is limited since the construction of this dataset employs some common methods. We are first to develop a data engine that can generate 200 thousand physically meaningful airfoils by both data collection and data synthesis, which is about 100 times larger and more diverse than previous airfoil datasets and includes all of supercritical airfoils, common airfoils and low-speed airfoils. Generally, the data engine also firstly combines the parametric perturbation methods and unconditional generation methods, i.e., diffusion models, while previous methods leverage parametric methods such as CST perturbation (described in L129-L137). Our data engine also leverages CFD software to filter meaningless airfoils. Such a large-scale dataset allows us to develop controllable airfoil generation and editing models that can be helpful to professional airfoil designers.
>
> Furthermore, we also have innovative strategies for using unconditional diffusion-based generative models to construct physically meaningful synthetic airfoils. Especially, when training the VAE in latent diffusion models, we discover that directly minimizing reconstruction loss and KL divergence loss will lead to model collapse, which means VAE reconstructs the same airfoil regardless of this input. Therefore, we dynamically weigh the original VAE loss (consisting of reconstruction loss and KL divergence loss) by training iterations, i.e., initially setting the weight of KL divergence loss to be 0 and then gradually increasing the weight to 0.1 while fixing the weight of reconstruction loss to be 1.

---

> ### Author Rebuttal · Authors · 2024-08-17
>
> **Q1:** Some technical jargon may be difficult for non-specialists.
>
> **A1:** Thank you for your feedback.
>
> First, we have provided explanations of some critical jargon about aerospace technology in our paper. Here are some examples:
>
> In Figure 4, we describe the specific meaning of geometric labels in airfoils (L154-L168).
>
> - Leading Edge Radius[L156]: As shown in Figure 4, the curvature at the front edge of an airfoil. It influences how air flows over the wing, affecting lift and drag characteristics.
>
> - Upper crest position[L156]: As shown in Figure 4, referring to the point on the upper surface where the height reaches its maximum, typically influencing airflow distribution and aerodynamic performance.
>
> In Appendix C, we also detail the CST method and the sampling strategy LHS:
>
> - CST (class-shape function transformation) [L529-L538]: A parametric approach used for designing and representing airfoil shapes and other aerodynamic surfaces. The CST method combines a class function for overall shape characteristics with a shape function to fine-tune geometry, enabling smooth and precise airfoil design.
>
> Following your suggestion, we will add detailed explanations of other jargon in the supplementary materials. Here are some examples:
>
> - CL (Lift Coefficient): A dimensionless number that represents the amount of lift generated by an airfoil relative to the air density, velocity, and wing area.
>
> - CD (Drag Coefficient): A dimensionless number that quantifies the drag or resistance an airfoil experiences as it moves through the air, relative to the same factors as CL.

---

> ### Author Response · Authors · 2024-08-27
>
> Dear Reviewer c69g,
>
> Our manuscript received a diverse set of scores and, as the only negative review, we are particularly interested in your feedback. We truly appreciate your valuable comments and insights. We are keen to know if our responses have adequately addressed your concerns. Your further feedback is highly valued, as it will aid us in improving our manuscript. Should you have any additional questions or need clarifications, please do not hesitate to inform us.
>
> Best regards,
>
> The Authors

---

### Official Review · Reviewer_jE49 · 2024-07-25
**Good and solid benchmark work; presented clearly easy to follow even by non-experts.**

**Rating:** 7
**Confidence:** 4
**Correctness:** Yes
**Clarity:** Yes

**Review:**

From the reviewer's perspective, this paper is well-written and publishable in terms of quality clarity originality, and significance.
I like the way that the problem is clearly defined what the inputs are what are the outputs, and the desired target/objectives.
And the dataset is documented.
One concern I would like to get an answer to is related to the problem setting itself: The dataset only includes airfoil design. Despite diverse sources of the data points, the entire dataset is still falls in airfoil design only which is somehow in a limited structure/topology.
I want to know how will this affect the ML model performance and affect the real airfoil inverse design process. Does it require additional optimization to achieve the final objective (aerodynamics, pressure, ...)

**Strengths:**

refer to the main review

**Additional Feedback:**

NA

**Documentation:**

Yes

**Opportunities For Improvement:**

Refer to main review

**Relation To Prior Work:**

Yes

**Summary And Contributions:**

The authors propose using generative methods for two primary tasks: conditional airfoil generation and airfoil editing.
The generation focuses on diversity, controllability, geometric quality, and aerodynamic quality.
Detailed data generation and collection are provided. and The data set covers a diverse source, making it a suitable evaluation benchmark effort.

---

> ### Author Rebuttal · Authors · 2024-08-17
>
> **Q1:** How the limited structure and topology of the airfoil-only dataset might impact (1) the ML model's performance, (2) the real airfoil inverse design process and (3) whether additional optimization steps are needed to achieve the final aerodynamic objectives?
>
> **A1:** Thanks for your review.
>
> (1) Effects on ML model performance. First, we would like to clarify the focus of this paper is airfoil design. The seemingly limited structure and topology result from that airfoils need to follow aerodynamics to ensure physical meaningfulness. The airfoils in our datasets are all physically meaningful, which ensures the generated designs remain within a physically meaningful space (success rate larger than 91%). Second, we would like to clarify that the structures in our AF-200K have been diverse (Please see Fig.1 in PDF-Rebuttal) in airfoils. For example, we are the first dataset that includes supercritical airfoils, common airfoils and low-speed airfoils. Therefore, our proposed baseline can generate various types of airfoils (supercritical airfoils, common airfoils and low-speed airfoils).
>
> (2) Effects on the real airfoil inverse design process. The authors of this paper have rich and practical experiences in real airfoil design. When the engineers design a new airfoil, we will search for some standard airfoil datasets, e.g., NACA, and select the airfoil whose physical and geometric properties are most similar to the requirements as the design initial points. Afterwards, the manual design will be done to achieve the exact requirements. With our controllable generation model, given any condition, the engineers can efficiently generate airfoils whose geometric properties are very close to the requirements. The generated airfoil then can be a good initial point for the professional engineer for further manual editing, which can significantly shorten about 60% time of designing a new airfoil in our practice. Furthermore, in the process of editing the initial airfoil, previously, engineers needed to drag every control point, which is very time-consuming. However, with our method of airfoil editing, engineers only need to drag one point, while other points will change accordingly to ensure a physically meaningful airfoil. This process can also shorten the editing time by 15%.
>
> (3) Additional Optimization. Yes. In practice, the engineers will implement some additional automatic optimizations, e.g., smoothing, to achieve final objectives, but normally require extensive manual design efforts. The engineers will further leverage CFD software to re-inspect pressure distributions and the aerodynamic parameters, i.e., the lift-to-drag ratio. Based on the results, the engineers will optimize the airfoil manually to enable the designed airfoil to conform to the requirements.

---

> ### Author Response · Authors · 2024-08-27
>
> Dear Reviewer jE49,
>
> We appreciate your valuable comments and insights on our manuscript! We are eager to know if our responses have effectively addressed your concerns. Your additional feedback is greatly valued, as it will assist us in enhancing our manuscript. Should you have any further questions or comments, please do not hesitate to reach out to us.
>
> Best regards,
>
> The Authors

---

### Official Review · Reviewer_LYMe · 2024-08-04
**a good and interesting article and a benchmark on the use of generative models for a relevant industrial task**

**Rating:** 7
**Confidence:** 4
**Correctness:** The claims made in the submission are…
**Clarity:** The paper is well written.

**Review:**

Airfoil modeling is a relevant industrial problem. An airfoil should be efficient for a number of R and M values, under different angles of attack, etc. Thus optimization of its design is a complex multi-criteria problem with a number of physical and geometrical constraints. The authors demonstrated how modern generative models can be used for airfoil generation and conditional airfoil generation, that can deliver an initial solution for airfoil inverse design problem.

Although, there exist papers about airfoil modelling based on generative models, it seems that this is the first paper that simultaneously delivers a big dataset, a number of unconditional/conditional generative models and their evaluation.

The paper is well written.

**Strengths:**

I personally like the paper. Airfoil modeling is a classic but still a very important industrial problem. And the authors demonstrated how modern ML techniques can facilitate its solution. Although the problem belongs to some specific applied domain, the main challenges of this problem and the main steps of the inverse design problem are the same in other engineering domains where we want to achieve an optimal object design. Based on the airfoil modeling benchmark, proposed in this paper, we can work out the principles of using modern generative models in engineering design problems.

**Additional Feedback:**

54-55: “…generation based on the single condition. The newly proposed airfoil editing task. Specifically, we currently support the editing …” - the sentence in the middle looks like some unfinished sentence.

81: “… the areodynamic performance, …” - misprint

In case of generative models with latent space not all regions of the latent space are useful - these regions can correspond to non-physical objects. Do the authors have any ideas how to control this?

114-115: “… such as lift-to-drag ratio or shar parameters, …” - what is “share parameters”?

127: “MAC(Commercial …)…” - a space is missing


248: “Where C_j is a binary variable …” - “where …”

**Documentation:**

There are sufficient detail on data collection etc.

**Ethics:**

I do not suspect any ethical concerns.

**Limitations:**

The authors adequately addressed the limitations, etc.

**Opportunities For Improvement:**

4.3. Baseline Methods. One of the basic steps in any predictive analytics task is to build a linear regression model to evaluate the main effects. In the work, the authors showed that even the standard linear PCA method allows for good quality profile generation. Accordingly, I would suggest using PCA as the basic method for comparison.

When performing controllable airfoil generation it is very important that physical properties of the generated airfoil fulfil certain restrictions. Why not to introduce PINN-like loss inside the loss function of a generative model to better guarantee fulfilment of the physical constraints?

In this kind of applications evaluation metrics are very important part of the whole process. In the paper “Comparison of Three Geometric Parameterisation Methods and Their Effect on Aerodynamic Optimisation”
https://www.researchgate.net/publication/290440849_Comparison_of_Three_Geometric_Parameterisation_Methods_and_Their_Effect_on_Aerodynamic_Optimisation
the authors proposed to evaluate airfoil parameterization methods (including a method based on nonlinear dimensionality reduction) the following characteristics:
- accuracy of reconstruction of airfoils. This metric is relevant for generaitve models with latent spaces (various autoencoders, etc.)
- success rate
- whether we can achieve diverse combinations between CL and CD
The authors used sucess rate, but not accuracy of reconstruction and diversity between CL and CD.
1) accuracy of reconstruction can be important for airfoil design optimization problems. Of course, the work proposed solving inverse design problems based on conditional generative models. However, they can provide only some initial solution which further should be optimized. Optimization can not be done in the initial space and some reduced order representation should be used. Dimensionality reduction methods with low-dimensional space are good candidates for such parameterization, e.g. auto-encoders (VAEs, etc.). Thus we should be сonfident in the airfoil, reconstructed from the optimal latent vector we find during the optimization process in the latent space.
2) diversity of combinations between CL and CD: CL and CD are to important global characteristics, which are often used as global criteria in airfoil design optimization problems. Thus it is important to have diverse set of CL and CD values for airfoils, represented by some generative model.

Are there any ideas how such issues (latent space fidelity, i.e. whether for some initial noise vector we have guarantees that the generative model generate an object description, airfoil, near the manifold of real physically meaningful airfoils; and accuracy of reconstruction) can be validated for generative models based on diffusions?

Let us assume that for some given physical constraints we know exactly the solution of the airfoil inverse design problem (e.g. by solving a classical optimization problem with corresponding constraints).
Then we can solve this airfoil inverse design problem using conditional generative models and get some approximate shape of the optimal airfoil. So in this case we can estimate efficiency of the conditional generative model by estimating a geometric proximity and aerodynamic proximity (say, difference between pressure distributions over the surfaces of the airfoils, calculated by some solver). I propose to characterize efficiency of the conditional generative models also using these characteristics.

**Relation To Prior Work:**

It is clearly discussed how this work differs from previous contributions.

**Summary And Contributions:**

The authors consider an airfoil modeling problem. They
- developed a large-scale and diverse airfoil dataset with respective geometrical and aerodynamic characteristics
- developed a number of generative models for airfoil generation
- developed a number of conditional generative models that can be used for airfoil inverse design
- developed a number of evaluation metrics to estimate airfoil quality

---

> ### Author Rebuttal · Authors · 2024-08-17
>
> **Q6:** Some clerical errors.
>
> **A6:** Thanks for the corrections. We will check the paper again in detail and correct errors in the final version.

---

> ### Author Rebuttal · Authors · 2024-08-17
>
> **Q5:** Evaluating conditional generative models by comparing their generated airfoil shapes and aerodynamic performance to solutions of a classical optimization problem. Use geometric and aerodynamic proximity as metrics.
>
> **A5:** Thanks for your suggestion. We choose PARSEC [37] as the classical parametric optimization method for airfoil inverse design. Given the 11 geometric parameters defined in L154-158, we generate airfoils by optimizing the PARSEC model. We also leverage the parameters as the conditions of our PK-DiT to generate the airfoil. We calculate the geometric proximity by averaging the pointwise L2-normed difference between airfoils generated by PARSEC and PK-DiT, respectively. We further leverage the CFD software, i.e., Xfoil solver, to calculate the flow map under 66 working conditions (L147-L153), and calculate the aerodynamic proximity (difference between pressure distributions over the surfaces of the airfoils) by the relative error metric defined in [B]. The experimental results in the Table below show that PK-Diff and PK-DiT can achieve good performance in both Aerodynamics and Geometric proximities.
>
> | Metric               | PK-DIFF | PK-DIT |
> |----------------------|---------|--------|
> | **Aerodynamics** ↓   |         |        |
> | CL (x0.01)           | 3.18    | 1.45   |
> | CD (x0.01)           | 2.39    | 1.33   |
> | **Geometric** ↓      |         |        |
> | $\sigma_1$           | 4.61    | 1.12   |
> | $\sigma_2$           | 3.46    | 3.23   |
> | $\sigma_3$           | 2.15    | 1.54   |
> | $\sigma_4$           | 277     | 105    |
> | $\sigma_5$           | 2.2     | 1.3    |
> | $\sigma_6$           | 1.93    | 1.15   |
> | $\sigma_7$           | 1030    | 979    |
> | $\sigma_8$           | 0.70    | 0.05   |
> | $\sigma_9$           | 0.11    | 0.05   |
> | $\sigma_{10}$        | 2.4     | 2.3    |
> | $\sigma_{11}$        | 3.1     | 2.4    |
> | $\bar{\sigma}$       | 120.6   | 99.7   |
>
> [B] Hidden fluid mechanics: Learning velocity and pressure fields from flow visualizations（Science）

---

> ### Author Rebuttal · Authors · 2024-08-17
>
> **Q4:** How can issues such as (1) latent space fidelity and (2) accuracy of reconstruction be validated in diffusion-based generative models?
>
> **A4:**
>
> (1) Although randomly selecting noise from Gaussian distribution for denoising has been a common practice in image and video generation, we have not found a paper that theoretically proves that any noise can generate meaningful samples.  Therefore, we design an experiment to empirically validate it. Specifically, we randomly sample 1000 noise from the Gaussian distribution given two different geometric conditions, i.e., one for reasonable geometric conditions and the other for unreasonable geometric conditions. We find that all 1000 noises can generate meaningful airfoils (L245-L250) given geometrically reasonable conditions, while all 1000 noises only generate meaningless airfoils given geometrically unreasonable conditions.
>
> (2) We leverage the airfoils in the testing set to validate the reconstruction error of VAEs in PK-Diff\&PK-DiT and notice that the reconstruction error is $7.61\times 10^{-8}$. Please refer to A3(1) for details.

---

> ### Author Rebuttal · Authors · 2024-08-17
>
> **Q3:** Highlighting the importance of including metrics like (1) accuracy of the reconstruction and (2) diversity in CL and CD, ...., to evaluate the generative model's performance.
>
> **A3:**  Thanks for your suggestion.
>
> (1) Following your suggestion, we calculate the reconstruction error of VAE in PK-Diff and PK-DiT for dimension reduction and notice that the reconstruction error is $7.61\times 10^{-8}$, which confirms that we can use the latent space generated by VAE for real airfoil optimization and decoding the real airfoils.  We will add the reconstruction error in L241 of the main text.
>
> (2) Due to the diversity of AFBench data, which contains various combinations of CL and CD, our model can generate the required airfoil under various reasonable combinations of CL (0.2 to 1.8) and CD (0.005 to 0.030). The results show that our method can generate airfoils that conform to the conditions and have diverse CL/CD combinations. We also would like to highlight that our diversity score (L234-L240) measures the geometric diversity of the generated airfoils given the same condition which will directly influence CL/CD diversity.

---

> ### Author Rebuttal · Authors · 2024-08-17
>
> **Q2:** Incorporating a PINN-like loss in the generative model to enforce physical constraints on the generated airfoils.
>
> **A2:** Thanks for your suggestion. We carefully consider your suggestions, but we think currently there are some challenges to incorporating PINNs in the airfoil inverse design.
>
> First, given any airfoil, PINNs can only generate meaningful flow fields after $10^5$ forward-backwards optimizations until network convergence. Therefore, it is impractical to incorporate the PINN-like loss into controllable airfoil generation for better physical constraints.
>
> Second, PINNs still fail to generate high-precison flow maps when its Raynold number is $10^5$ (cruise phase in flight) [A] in our paper.
> Despite some technical challenges, we believe that adding PINN-like loss to airfoil inverse design should be carefully explored in the future.
>
>  [A] Solving high-dimensional parametric engineering problems for inviscid flow around airfoils based on physics-informed neural networks (Journal of computational physics)
>
> We propose an alternative approach to regularize the generated airfoil to fulfil certain physical constraints. We train a surrogate model to estimate the lift-to-drag ratio (CL/CD) of the reconstructed airfoils, and use this estimation as part of the loss function. Then, we use CL/CD as a conditional variable in our inverse design process. Our experimental results illustrate the airfoils generated by PK-VAE have more consistent aerodynamical properties with aerodynamical conditions by the errors between CL/CD of generated airfoils and required CL/CD (CVAE: $7.84\times 10^{-2}$, CGAN: $9.17\times 10^{-2}$, PK-DIFF: $3.18\times 10^{-2}$, PK-DIT: $1.45\times 10^{-2}$).

---

> ### Author Rebuttal · Authors · 2024-08-17
>
> **Q1:** Using PCA as a baseline method for comparison in airfoil design.
>
> **A1:**  Thanks for your suggestion.
>
> First, we would clarify the PCA method cited in L486 can only tackle unconditional airfoil generation instead of conditional airfoil generation in our paper. Specifically, it leverages PCA to compress every airfoil to 3-dimensional latent features, add noise to the latent features, and decode them into new airfoils.
>
> To enable conditional airfoil generation by PCA, we slightly modify the original PCA-based method to enable controllable airfoil generation. Specifically, in the training stage, we use the samples in the entire training set (AF-200K) to calculate the projection matrix, where the hidden dimension is set to 10. In the inference stage, we search for 10 training samples closest to the given conditions and use the trained PCA to obtain their latent features. Then we average these 10 latent features to estimate the latent of the target airfoil. To enable the diversity of airfoil generation, we randomly add some Gaussian noise to the estimated target latent features to decode the airfoil by inverse PCA transformation. As shown in the Table below, it can be found that the naive PCA results are inferior to our proposed conditional generation model, i.e., PK-Diff, and PK-DiT, with worse proximity to the conditions, lower smooth, and lower diversity.
>
> | Metric               | PCA   | CVAE  | CGAN  | PK-DIFF | PK-DIT |
> |----------------------|-------|-------|-------|---------|--------|
> | **Label Error** ↓    |       |       |       |         |        |
> | $\sigma_1$           | 14.3  | 7.29  | 10.7  | 4.61    | 1.12   |
> | $\sigma_2$           | 21.1  | 5.25  | 8.50  | 3.46    | 3.23   |
> | $\sigma_3$           | 14.2  | 3.52  | 5.44  | 2.15    | 1.54   |
> | $\sigma_4$           | 8900  | 1590  | 2320  | 277     | 105    |
> | $\sigma_5$           | 56.9  | 9.9   | 14.3  | 2.2     | 1.3    |
> | $\sigma_6$           | 37.7  | 9.55  | 13.7  | 1.93    | 1.15   |
> | $\sigma_7$           | 10100 | 2900  | 5960  | 1030    | 979    |
> | $\sigma_8$           | 10.87 | 1.91  | 2.53  | 0.70    | 0.05   |
> | $\sigma_9$           | 11.77 | 1.53  | 2.23  | 0.70    | 0.05   |
> | $\sigma_{10}$        | 31.1  | 4.6   | 5.3   | 2.4     | 2.3    |
> | $\sigma_{11}$        | 60.5  | 10.4  | 12.9  | 3.1     | 2.4    |
> | $\bar{\sigma}$       | 1750.7| 413.1 | 759.6 | 120.6   | 99.7   |
> | **Diversity** ↑      |       |       |       |         |        |
> | $\mathcal{D}$        | -548.7| -155.4| -120.5| -101.3  | -93.2  |
> | **Smoothness** ↓     |       |       |       |         |        |
> | $\mathcal{M}$        | 51.39 | 7.09  | 7.31  | 1.52    | 1.04   |

---

> ### Author Response · Authors · 2024-08-27
>
> Dear Reviewer LYMe,
>
> Thank you for providing your valuable comments and insights on our manuscript. We are keen to learn if our responses have successfully addressed your concerns. Your further feedback is highly appreciated, as it will help us improve our manuscript. If you have any additional questions or comments, please do not hesitate to contact us.
>
> Best regards,
>
> The Authors

---

### Decision · Program_Chairs · 2024-09-26

**Decision:**

Accept (Poster)

**Comment:**

The papers got mostly positive ratings with one negative side: Accep, Accep, Marginally below the accept bar, Clear accept.

The AC believes the authors’ rebuttal addresses the concerns of reviewers mostly. The AC also supports accepting the paper and strongly suggests the authors to follow the reviewers’ feedback in the camera-ready version.